# Molecular Detection of Different Species of *Cryptosporidium* in Snakes from Surinam and Indonesia

**DOI:** 10.3390/ani15111556

**Published:** 2025-05-26

**Authors:** Magdaléna Polláková, Monika Sučik, Vladimír Petrilla

**Affiliations:** 1Department of Biology and Physiology, University of Veterinary Medicine and Pharmacy in Košice, Komenského 73, 041 81 Košice, Slovakia; magdalena.pollakova@uvlf.sk (M.P.); vladimir.petrilla@uvlf.sk (V.P.); 2Zoological Department, Zoological Garden Košice, Široká 31, Kavečany, 040 06 Košice, Slovakia

**Keywords:** *Cryptosporidium* spp., reptiles, snakes, PCR, zoonotic diseases

## Abstract

The global trade and husbandry of exotic reptiles have expanded significantly, raising concerns about associated health and ecological risks. This study examined the presence of *Cryptosporidium* spp., a microscopic parasite, in feces of wild-caught snakes from Suriname and Indonesia before their introduction into private collections. Fecal samples from 40 individuals were analyzed, revealing cryptosporidial oocysts in six cases. Notably, the detected species—*C. hominis*, *C. parvum*, and *C. tyzzeri*—are primarily associated with humans and mammals rather than reptiles, indicating potential zoonotic implications and a possible risk of transmission to people who handle these animals. This finding highlights the importance of regular health screening for exotic reptiles to prevent the spread of infections. Additionally, releasing or relocating infected animals could introduce these parasites into new environments, potentially affecting local wildlife and disrupting ecosystems. Ensuring proper veterinary monitoring and hygiene measures is essential to reduce these risks. This study emphasizes the need for responsible handling of wild-caught reptiles to protect both human and animal health while maintaining ecological balance.

## 1. Introduction

The presence of parasitic protozoans from the genus *Cryptosporidium* in reptiles, especially snakes, has already been described by different authors, who highlighted the zoonotic potential, chronic nature of the infection as well as the possible cross-transmission of pathogens between different host species, e.g., wild animals, domestic animals, and humans [1].

Out of 38 currently identified *Cryptosporidium* species, only 4 species are known to infect reptiles (*Cryptosporidium serpentis*, *Cryptosporidium testudines*, *Cryptosporidium varanii* (*saurophilum*), and *Cryptosporidium ducismarci*), from which only 2 species were found to cause diseases in snakes [2,3,4]. The diagnosis of the presence of pathogenic cryptosporidial oocysts in the organism of reptiles as well as the determination of an ongoing cryptosporidial infection have certain limitations that should be addressed.

One of the possible problems of diagnosis is the relatively small size of oocysts and low number of oocysts excreted during the subclinical stage of cryptosporidial infection, which may lead to false-negative results of testing. On the other hand, conventional morphologic identification of oocysts through microscopic examination using a stool smear is not considered to be a reliable diagnostic method either, as it is difficult to differentiate pathogenic oocysts from those which just pass through the gastrointestinal tract, and a potential problem with false-positives may occur due to the cross-reaction with other species of *Cryptosporidium* (e.g., *C. parvum*, *C. muris*) whose oocysts are eliminated at the time of feeding. Therefore, euthanizing reptiles tested positive for the presence of oocysts is not a recommended prevention strategy, as this practice might lead to the killing of uninfected animals [5,6,7].

Certain conditions of breeding facilities, such as sanitary management, ambient temperature, and high humidity, among other factors, can maintain and prolong the viability of oocysts. The inactivation of oocysts using disinfectants was not proved to be efficient, and the usage of traditional anticoccidials also often does not produce the desired results. Limited treatment options for cryptosporidiosis require the introduction of husbandry approaches with an emphasis on preventive measures and strict hygiene practices. Parasitological monitoring of new individuals before their introduction into collections as well as the testing of specimens in captive populations should be periodically carried out, even among clinically healthy animals. A study carried out in the premises of the Barcelona Zoo considers the individuals with chronic cryptosporidial infection to be the primary transmission route and a source of protozoa and cause of re-infections in enclosures [8].

Thorough monitoring of the collection and identification of infected individuals enables the isolation of suspicious animals and thus prevents the transmission of infections in captive environment. Moreover, screening for protozoan diseases ensures the preservation of the breeding quality and genetic lines of snakes [1,9].

The aim of our study was to monitor the health status of wild-caught snakes collected in Indonesia for the purpose of placement in private breeding collections. Many of these facilities possess high-value specimens of expensive snake species, and it is therefore necessary to verify parasitic status before introducing new individuals. At the same time, factors such as stress and discomfort during transportation must be considered, given that they can contribute to the outbreak of the initially subclinical infection.

After the snakes were caught in the wild, their feces were collected for further veterinary analysis focused on the presence of cryptosporidial oocysts.

In our study, we analyzed the presence of *Cryptosporidium* spp. in the feces of several wild-caught snake species, namely *Boa constrictor*, *Corallus caninus*, and *Corallus hortulanus* from Surinam, and *Boiga cynodon*, *Boiga dendrophila* spp. *gemmicincta*, *Boiga irregularis*, *Chrysopelea paradisi*, *Gonyosoma janseni*, *Oligodon octolineatus*, and *Trimeresurus insularis* from Indonesia (Table 1).

## 2. Materials and Methods

### 2.1. Collection of Samples

Stool samples were collected from 10 species of snakes (40 snakes) originating from Surinam and Indonesia. All responsibilities related to the handling of snakes, including animal welfare, sampling procedures, antiparasitic treatments, and health-status monitoring, were fully covered by the company MD—REPTILES, s.r.o. The biological material, including fecal samples and mice used for feeding the snakes, was provided by MD—REPTILES, s.r.o. solely for the purposes of scientific research.

The droppings were collected from February to May after their arrival in Slovakia. During their stay in Slovakia, they were fed mice, which we tested for the presence of cryptosporidial oocysts and were negative. None of the animals showed any gastrointestinal clinical signs. After the collection itself, the samples were kept in a freezer until analysis.

Based on the information provided to us, none of the snakes exhibited any overt signs of disease or physical abnormalities, and no data regarding mortality were reported.

### 2.2. DNA Extraction

Feces samples were applied to microtubes containing glass (0.5 mm) and zirconium beads (1.0 mm), ensuring better mechanical disruption of cryptosporidial oocysts. After pouring in the lysis solution, they were homogenized 2 × 45 s at 6500 rpm using a Precellys 24 device (Berlin Technologies, GmbH, Berlin, Germany). DNA was isolated according to the manufacturer’s instructions using the AmpliSens^®^ DNA-sorb-B isolation kit (InterLabService, Moscow, Russia), which is designed for stool processing. If the samples were not processed immediately, they were stored in a freezer at −20 °C.

### 2.3. PCR Amplification

To amplify the 60 kDa glycoprotein gene (*gp60*) of *Cryptosporidium* spp., Nested PCR was used according to Xiao (2010) [21]. Master Mix—45 μL (Solis BioDyne, Tartu, Estonia) contained 5 U Taq DNA polymerase (FIREpol), 0.1 μM primers GP_F1 (5′-ATGAGATTGTCGCTCATTATC-3′), GP_R1 (5′-TTACAACACGAATAAGGCTGC-3′), GP_F2 (5′-GCCGTTCCACTCAGAGGAACC-3′), GP_R2 (5′-CACATTACAAATGAAGTGCCGC-3′). Reactions were performed in XP Thermal Cycler Blocks, and the program consisted of incubation at 95 °C for 5 min, 30 cycles of denaturation at 95 °C for 30 s, annealing at 54/58 °C for 45 s, termination at 72 °C for 1.5 min and final polymerization at 72 °C for 7 min. We then analyzed the final 450 bp products (for primers targeting the *gp60* gene) on a 1.5% agarose gel stained with GoodView-Nucleic Acid Stain in TAE buffer. The sequencing service verified positive samples using the Sanger sequencing method, and the final sequences were compared to homologous sequences deposited in GenBank using ElasticBLAST 1.4.0.

## 3. Results

Successful cryptosporidial species identification was achieved through sequence analysis in 6 of 40 samples (Table 2). Four positive snakes came from Indonesia and two from Surinam. *C. hominis* was detected in two snakes of the species *Boiga gemmicincta*, *C. parvum* was detected in *Oligodon octolineatus* and *Trimeresurus insularis*, and *C. tyzzeri* was found in snakes originating from Surinam, namely *Corallus hortulanus* and *Corallus caninus*.

## 4. Discussion

Cryptosporidial infections in snakes can lead to a range of health issues, including gastrointestinal disorders, which result in a general state of poor health. Affected snakes may exhibit signs of dehydration, lethargy, and loss of appetite. In more severe cases, prolonged infection can lead to weight loss, emaciation, and an overall debilitated appearance. Respiratory issues may also be observed, although less frequently. These clinical signs are often accompanied by a general weakness, which can be fatal if not addressed. Cryptosporidiosis is typically a condition that resolves on its own in healthy animals, but it can pose a life-threatening risk to immuno-compromised individuals and juveniles [22].

When establishing a definitive diagnosis of cryptosporidial infection and subsequently implementing effective treatment, it is essential to also consider the fact that infected prey may serve as a source of oocysts, which can passively transit through the reptile’s gastrointestinal tract without causing an actual infection [23]. *Cryptosporidium* species that have been isolated from reptilian feces include *Cryptosporidium baileyi*, *C. muris*, *C. parvum* mouse genotype, and *C. parvum* bovine genotype. The authors of the study report that no infections in humans had been linked with reptilian *Cryptosporidium* species (*C. serpentis*, *C. testudines*, *C. varanii*, and *C. ducismarci*) to date [1].

In the case of *Cryptosporidium parvum* species, one report suggested that *C. parvum* is not transmissible to reptiles, fish or amphibians. Authors state that under certain circumstances, e.g., the ingestion of *C. parvum*-infected prey, these animals may possibly disseminate *C. parvum* oocysts in the environment [24]. For this reason, we decided to test the mice that were used to feed the snakes after their capture, for the presence of *C. parvum* oocysts in the mice’s gastrointestinal tract. The results of all the samples obtained from mice were negative, which suggests that the presence of cryptosporidial oocysts in the feces of the tested snakes was not caused by the administration of contaminated food.

The occurrence of the obligatory parasite *Cryptosporidium* spp. has been studied by several scientists from different parts of the world. Of the listed species (as can be seen from Table 3), *C. serpentis* predominates, which infects most wild snakes but is also easily transmitted to snakes kept in captivity. In comparison with our results, Lobão et al. from Rio de Janeiro [25] detected *C. parvum* and *C. tyzzeri* in equal amounts in captive snakes. However, we have not found any author who has so far detected *C. hominis* in snakes, as we confirmed in the species *Boiga dendrophila* spp. *gemmicincta*, which indicates a greater risk of zoonotic potential.

There are different opinions among authors about the health risks that *C. parvum* possess for human health. In the review published by Šlapeta [35], this parasite species can be considered of minor public health significance, as the author stated that there was only one recorded case of *C. parvum* in humans to date. However, a more recent study conducted by Ryan et al. [36] stated that out of all currently recognized *Cryptosporidium* species, 19 species and 4 genotypes have been reported in humans, with *C. hominis*, *C. parvum*, *C. meleagridis*, *C. canis* and *C. felis* being the most prevalent. Given these conflicting perspectives, we recommend exercising caution and adhering to safety measures when collecting and analyzing samples, to prevent the spread of oocysts and contamination. Notably, two of our samples tested positive for the presence of *Cryptosporidium parvum*: one obtained from an *Oligodon octolineatus* specimen caught in Indonesia and another from a *Trimeresurus insularis* specimen, also caught in Indonesia.

## 5. Conclusions

Employing modern molecular methods, we conducted a screening for the presence of cryptosporidial oocysts in the feces of snakes captured from the wild, intended for sale. Our results demonstrate the effectiveness of these molecular techniques in detecting cryptosporidial infections, even in asymptomatic individuals, highlighting the importance of screening wild-caught reptiles for parasitic diseases before they enter trade and are placed into private collections. It is also important to emphasize that, since a single negative PCR result based on one-time fecal testing may yield false negative results, it is essential that breeders and all individuals in contact with such animals remain cautious.

Our approach provides a valuable tool for ensuring the health and welfare of reptiles in captivity, offering an early detection system for potential health risks. For all snakes involved in this study, antiparasitics were recommended as a preventive measure against cryptosporidial infections, as well as other diseases, such as helminthiasis. Although no snake-specific *Cryptosporidium* species were found in the analyzed samples, the identified species may still pose a risk to humans, which should be considered when working with wild reptiles.

## Figures and Tables

**Table 1 animals-15-01556-t001:** General information about the studied snake species.

Snake Species	Number of Snakes Examined	Country of Origin	Feeding Ecology	The Site of Snake Collection	References
*Boa constrictor* (Common boa)	2	Central and South America	Large lizards, small- or moderate-sized birds, opossums, bats, mongooses, rats, and squirrels	Surinam	[10]
*Boiga cynodon* (Dog-toothed cat snake)	2	Asian palm plantations	Small birds of all sorts and bird eggs, lizards and small bats, chickens, eggs; captive animals: mice	Indonesia	[11,12]
*Boiga gemmicincta* (Mangrove snake)	2	Sulawesi and surrounding islands	Birds, rodents, reptiles (including lizards and other snakes), amphibians (frogs), fish	Indonesia	[13]
*Boiga irregularis* (Brown Tree Snake)	3	Throughout subtropical and tropical eastern and northern Australia	Large variety of vertebrate prey items (Mammalia, Aves, Reptilia, Amphibia); feeds on carrion	Indonesia	[14]
*Genus Corallus* (Tree boas)	2	From southeastern Guatemala, in Central America, to southeastern Brazil, in South America	Lizards, birds, and small mammals (marsupials, rodents, and bats)	Surinam	[15]
*Chrysopelea paradisi* (Paradise flying snake)	7	Southeast Asia, India, western Indonesia, Philippines	Tree-dwelling, lizards, birds, rodents, and bats	Indonesia	[16]
*Gonyosoma janseni* (Black-tailed Ratsnake)	5	Sulawesi (Indonesia), adjacent island Salayar	Rodents, birds, hatchlings and possibly lizards	Indonesia	[17]
*Oligodon octolineatus* (Striped Kukri Snake)	2	Peninsular Malaysia, Singapore, Borneo, Brunei and the Indonesian islands of Sumatra, Bangka, Bali, Java, Sulawesi, Philippines	Frogs, lizards, other snakes and eggs of frogs, reptiles and birds	Indonesia	[18]
*Trimeresurus insularis* (Indonesian pit viper)	15	Indonesia and Timor-Leste	Rodents, lizards, and small birds	Indonesia	[19,20]

**Table 2 animals-15-01556-t002:** The results of the analysis of the examined samples confirmed by sequencing.

Snake Species	Country of Origin	Positivity/ Negativity	Number of Positive Samples of the Given Species	*Cryptosporidium* spp.	GenBank Accession Numbers	Allelic Family
*B. constrictor*	Surinam	−	0	−		
*C. caninus*	Surinam	+	1	*C. tyzzeri*	PV656777	IXb
*C. hortulanus*	Surinam	+	1	*C. tyzzeri*	PV656778	IXb
*B. cynodon*	Indonesia	−	0	−		
*B. dendrophila* spp. *gemmicincta*	Indonesia	+	2	*C. hominis*	PV656780 PV656781	Ia Ia
*B. irregularis*	Indonesia	−	0	−		
*Ch. paradisi*	Indonesia	−	0	−		
*G. janseni*	Indonesia	−	0	−		
*O. octolineatus*	Indonesia	+	1	*C. parvum*	PV656779	IIa

**Table 3 animals-15-01556-t003:** Overview of the occurrence of *cryptosporidium* spp. in snakes worldwide.

Country of Sample Analysis	Living	Species of Positive Snakes	Parasite Detected	References
Rio de Janeiro	In captivity	*Bothrops* spp., *Crotalus durissus*, *Lachesis muta*	*C. parvum*, *C. tyzzeri*	[25]
Italy	In captivity	*E. quatuorlineata*, *H. viridiflavus*	*Cryptosporidium* spp.—too short for reliable species identification	[26]
Florida	Wildlife, In captivity	*Drymarchon couperi*	*C. serpentis*	[27]
Tehran	In captivity	*Macrovipera lebetina obtusa*, *Echis carinatus sochureki*, *Ophiophagus hannah*	*Cryptosporidium* spp.	[28]
China	In captivity	pet snake species, *Elaphe guttata*, *Elaphe obsoleta*, *Pituophis melanoleucus*, *Thamnophis sirtalis*, *Lampropeltis getulus*, *Heterodon nasicus*	*C. serpentis*, *C. varanii*	[29,30,31]
Louisiana	In captivity	*Pituophis ruthveni*	*C. serpentis*	[32]
Minas Gerais (southeast Brazil)	Wildlife	*Crotalus durissus terrificus*	*Cryptosporidium* spp.	[33]
Germany	In captivity	pet snake species	*Cryptosporidium* spp.	[34]

## Data Availability

The original contributions presented in this study are included in the article. Further inquiries can be directed to the corresponding author.

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
