# Peer review of "Molecular Detection of Different Species of Cryptosporidium in Snakes from Surinam and Indonesia"

_animals, 2025, doi:10.3390/ani15111556_

Round 1

Reviewer 1 Report

Comments and Suggestions for Authors

Dear authors,

I have detected numerous issues that must be addressed in order to proceed with the review process. I list my suggestions below:

L12, 14, 30, 32, 35, 37, 40, 43, 48, 51, 52, 55, 61, 111, 133, 144, 147, 158, 166, Table 3 (several times), 182, 184, 189, 190: Cryptosporidium should be italicized.
L43-92: The entire section includes only a single bibliographic reference (Brener et al., 2022). This is entirely unacceptable. The authors present information from other studies without citing them. It is essential to increase the number of references and to be more precise when attributing content.
L73-75: An example of the above is found here, where a study conducted in Barcelona (Spain) is mentioned, while the cited study (Brener et al., 2022) is limited to Brazil.
L76-79: This paragraph should include at least one reference.
L86-92 and Table 1: This content corresponds to Materials and Methods, not to the Introduction.
Table 1: The “Supplementary information” column is irrelevant. It does not provide useful information for the study.
Table 1, Boiga irregularis: The fact that it feeds on carrion is part of its feeding ecology and should be included in that column.
Table 1 (new column): Since this table should be placed in the Materials and Methods section, it would be appropriate and beneficial to include a new column indicating the number of specimens sampled from each species (or genus).
L96-101: Why did the animals arrive in Slovakia? Was it due to the international trade of animals or for another reason? Was Slovakia their final destination or just a temporary one? (During their stay in Slovakia...).
L112-122: The authors must include the references corresponding to the amplification protocols and reagents used.
L130 (Table 2): Why is the “Allelic family” column highlighted in yellow?
L142-144: References should be provided to support these statements.
L150, 152: The expression “lower vertebrate” seems outdated (species should not be classified as superior or inferior). Could the resistance to C. parvum be due to their poikilothermic condition?
L188: Which antiparasitic treatments? Please justify your response, considering the comments made in lines 68-69.

It is essential that the authors improve their text, revise the content of each section accordingly, and provide proper references to support the information presented, in order to proceed with the review process.

Author Response

Comment: L12, 14, 30, 32, 35, 37, 40, 43, 48, 51, 52, 55, 61, 111, 133, 144, 147, 158, 166, Table 3 (several times), 182, 184, 189, 190: Cryptosporidium should be italicized.

Answer: We have italicized the term Cryptosporidium throughout the entire manuscript. Please find the corrected version of our manuscript attached.

L43-92: The entire section includes only a single bibliographic reference (Brener et al., 2022). This is entirely unacceptable. The authors present information from other studies without citing them. It is essential to increase the number of references and to be more precise when attributing content.

Answer: We apologize for the situation. The 1.Introduction section has been carefully revised, and we have included all original references, not just the one article that cited them. A total of eight references have been included to 1.Introduction section.

Revisions have been made in both the 1.Introduction and References sections. Kindly see the corrected manuscript.

L73-75: An example of the above is found here, where a study conducted in Barcelona (Spain) is mentioned, while the cited study (Brener et al., 2022) is limited to Brazil.

Answer: As mentioned in our previous comment, the Introduction section has been improved, and we have included all original references, not just the one article that cited them. Regarding the study conducted in Barcelona, we have added the following reference:

Gracenea, M.; Gomez, M.S.; Torres, J.; Carne, E.; Fernández-Morán, J. Transmission dynamics of Cryptosporidium in primates and herbivores at the Barcelona zoo: A long-term study. Vet. Parasitol. 2002, vol. 104, p.19–26.

Revisions have been made in both the 1.Introduction and References sections. Kindly see the corrected manuscript.

L76-79: This paragraph should include at least one reference.

Answer: The paragraph in question is partly our own conclusion, drawn based on the article by Brener et al., 2022. Therefore, we have included Brener et al., 2022 as a reference, along with an article cited by Brener et al. in this context:
Samuel, W.M.; Pybus, M.J.; Kocan, A.A. Parasitic Diseases of Wild Mammals, 2nd ed.; Iowa State University Press: Ames, IA, USA, 2001.

Revisions have been made in both the 1.Introduction and References sections. Kindly see the corrected manuscript.

L86-92 and Table 1: This content corresponds to Materials and Methods, not to the Introduction.

Answer: Table 1 has been moved to section 2. Materials and Methods, specifically to subsection 2.1. Collection of samples. Please see Table 1 in the corrected manuscript.

Table 1: The “Supplementary information” column is irrelevant. It does not provide useful information for the study.

Answer: The Supplementary informationcolumn has been removed from Table 1, as this information is not relevant to the study. Kindly refer to Table 1 in the corrected version of the manuscript.

Table 1, Boiga irregularis: The fact that it feeds on carrion is part of its feeding ecology and should be included in that column.

Answer: We have included information that Boiga irregularis feeds on carrion in the 'Feeding ecology' column. Kindly refer to Table 1 in the corrected version of the manuscript.

Table 1 (new column): Since this table should be placed in the Materials and Methods section, it would be appropriate and beneficial to include a new column indicating the number of specimens sampled from each species (or genus).

Answer: We added a column to the first table with the number of snakes examined.

L96-101: Why did the animals arrive in Slovakia? Was it due to the international trade of animals or for another reason? Was Slovakia their final destination or just a temporary one? (During their stay in Slovakia...).

Answer: All responsibilities related to the handling of snakes, including animal welfare, sampling procedures, antiparasitic treatments, and health status monitoring, were fully covered by the company MD - REPTILES, s.r.o.. As we mentioned in the 1.Introduction section of our manuscript, the aim of our study was to experimentally examine the presence of cryptosporidial oocysts in the feces of wild-caught snakes collected in Indonesia, intended for placement in private breeding collections. The droppings were collected from February to May, after the snakes' arrival and during their temporary stay in Slovakia. Slovakia served only as a transit location, as the company is based there and the snakes were subsequently sold for placement in private breeding collections in EU. Based on your comment, as well as those of the other reviewers, we have decided to include additional information about the company MD - REPTILES, s.r.o. in our manuscript. Kindly see Lines 96-100 in the corrected version of the manuscript.

L112-122: The authors must include the references corresponding to the amplification protocols and reagents used.

Answer: The reference has been added.

L130 (Table 2): Why is the “Allelic family” column highlighted in yellow?

Answer: The “Allelic family”  column was highlighted because this was the improvement required before our paper could be considered for peer review. We reviewed the academic editor's comments and resubmitted the revised manuscript with the highlighted correction. However, considering the current stage of the review process, the highlighting is no longer required, and thus we have removed it. Kindly see Table 2 in corrected version of manuscript.

L142-144: References should be provided to support these statements.

Answer: We have included a reference to support these statements in our manuscript:
Karasawa, A.S.M.; da Silva, R.J.; Mascarini, L.M.; Barrella, T.H.; de Magalhães Lopes, C.A. Occurrence of Cryptosporidium (Apicomplexa, Cryptosporidiidae) in Crotalus durissus terrificus (Serpentes, Viperidae) in Brazil. Memórias Inst. Oswaldo Cruz 2002, vol. 97, p.779–781.

Kindly see Line 152 in the corrected version of our manuscript.

L150, 152: The expression “lower vertebrate” seems outdated (species should not be classified as superior or inferior).

Answer: The term "lower vertebrates" was used based on the wording of the original literature we referenced. However, as this term is now considered outdated, we have slightly rephrased the relevant sentences. Kindly see Lines 158-159 in the corrected version of our manuscript.

Could the resistance to C. parvum be due to their poikilothermic condition?

Answer: So far, we have not encountered research that addresses the links between resistance to C. parvum and the poikilothermic state of snakes.

L188: Which antiparasitic treatments? Please justify your response, considering the comments made in lines 68-69.

Answer: Based on the provided information, we are unable to accurately determine the treatments the snakes underwent. The company responsible for the import and sale of these individuals, according to the information available to us, recommends treatment upon sale; however, the specific procedures followed by individual breeders are unknown to us and cannot be verified. These details have been specified in our manuscript (Lines 197-199). Kindly see the corrected version.

It is essential that the authors improve their text, revise the content of each section accordingly, and provide proper references to support the information presented, in order to proceed with the review process.

Reviewer 2 Report

Comments and Suggestions for Authors

This study reports the presence of C. parvum, C. hominis, and C. tyzzeri in snakes from Surinam and Indonesia. Among them, C. hominis is the first report in snakes world wild.

Here is my main concern:

In Cryptosporidium genotyping and subtyping studies, species should be detected by the small subunit (SSU) rRNA gene, while subtyping for some species would be performed by sequence analysis of the GP60 gene. However, the author used the nested PCR targeting the GP60 gene to confirm the Cryptosporidium species and subtype family, and only presented the data in a table. Please reanalyze the samples with SSU rRNA and gp60, provide detailed data with reference sequences, including sequence similarity and phylogenetic tree, etc.

Author Response

Answer: We understand your request, but due to current technical conditions, it is impossible for us to re-examine the samples. We have added the requested sequences to table number 2, and we can add a possible phylogenetic tree within approximately two weeks, if necessary for the publication of this article.

Reviewer 3 Report

Comments and Suggestions for Authors

Dear authors

There are three main points to be clarified here, as follows

-. My main concern related to the animal welfare in this case is the legal permit/s to caught and handle these snakes in the wild and to keep in captivity, particularly involving a zoologic institution. All this information of permits for legal trading must de added in the manuscript to avoid any issue related to wildlife illegal trade. Another related point to be clarified here is if these wild-caught snakes were in a quarantine section immediately to the arrival to destination, in order to avoid the entrance of any exotic and catastrophic disease to the facilities of the zoologic institution.

-. Criteria for fecal sampling and PCR testing against Cryptosporidium spp should be clarified here. According to the data included in the manuscript, one sample of each of the snakes studied here were tested once against this protozoal microorganism. Please consider that the shedding of this parasite could be intermittent by carrier or infected snakes. Thus, regular testing is necessary to detect true positive animals. In this case, negative PCR result after one fecal testing could provide false negative results.

-. Cryptosporidium spp. is related to proliferative gastritis and rapid decreasing of body condition in snakes in captivity, particularly. Because no postmortem studies were performed here, the authors are recommended to include information about body condition of each of snakes tested here, and if another common disease (e.g. inclusion body disease) was tested in the way to check the health status of these wild-caught snakes.

Also, minor editing is suggested, as follows

-. The authors are recommended to include all common names of the tested snakes, together with scientific names of each one.

-. The authors are recommended to unify the naming of the protozoal parasite studied here as "Cryptosporidium" or "Cryptosporidium spp." in all manuscript.

-. The authors are recommended to include reference/s for the molecular study, if the the set of primers and PCR cycling conditions were obtained from previous published papers.

Author Response

Dear authors

There are three main points to be clarified here, as follows

-. My main concern related to the animal welfare in this case is the legal permit/s to caught and handle these snakes in the wild and to keep in captivity, particularly involving a zoologic institution. All this information of permits for legal trading must de added in the manuscript to avoid any issue related to wildlife illegal trade. Another related point to be clarified here is if these wild-caught snakes were in a quarantine section immediately to the arrival to destination, in order to avoid the entrance of any exotic and catastrophic disease to the facilities of the zoologic institution.

Answer: All responsibilities related to the handling of snakes, including animal welfare, sampling procedures, antiparasitic treatments, and health status monitoring, were fully covered by the company MD - REPTILES, s.r.o. . At our institution University of Veterinary Medicine and Pharmacy, department of Biology, which focuses on the screening of Cryptosporidium infections, we do not have direct contact with live animals. Only fecal and dietary samples were provided to us exclusively for research purposes. The snake feces were supplied by company MD - REPTILES, s.r.o. specifically for our scientific study, and this type of Cryptosporidium testing is not a part of their routine practice. Therefore, we are not in a position to provide detailed information to questions regarding the handling of the animals themselves. According to the information provided, wild-caught snakes collected in Indonesia were intended for placement in private breeding collections in EU. This information was already included in the initial version of our manuscript, specifically in section 1. Introduction - sixth paragraph. Given the long-standing operation and experience of the company MD - REPTILES, s.r.o., we assume that their procedures are in compliance with applicable laws and guidelines, and that their measures are sufficient to prevent the spread of exotic and catastrophic diseases to the facilities of the zoological institution. We have included this clarification in the corrected version of the manuscript. Kindly see Lines 96-100.

-. Criteria for fecal sampling and PCR testing against Cryptosporidium spp should be clarified here. According to the data included in the manuscript, one sample of each of the snakes studied here were tested once against this protozoal microorganism. Please consider that the shedding of this parasite could be intermittent by carrier or infected snakes. Thus, regular testing is necessary to detect true positive animals. In this case, negative PCR result after one fecal testing could provide false negative results.

Answer: As mentioned in previous answer, all responsibilities related to the handling of snakes, including animal welfare, sampling procedures, antiparasitic treatments, and health status monitoring, were fully covered by the company MD - REPTILES, s.r.o. . Only fecal and dietary samples were provided to us exclusively for research purposes. The snake feces were supplied by company MD - REPTILES, s.r.o. specifically for our scientific study, and this type of Cryptosporidium testing is not a part of their routine practice. The samples were not tested immediately upon arrival but were stored at –80°C for a certain period before processing. Both sample preparation and analysis were carried out over the course of several months. The samples were provided to us for research purposes only, and the company MD - REPTILES, s.r.o. did not request any results from the analyses, which were intended to serve an experimental rather than a diagnostic function. Given these circumstances, it is no longer possible to obtain additional samples from the same individuals or to conduct follow-up analyses. Regular testing would only be possible if company MD - REPTILES, s.r.o. expressed interest in such monitoring. As the owners of the animals, they are solely responsible for any health screening, and without their explicit consent, it is neither possible to obtain any biological material nor to intervene in their routine practices in any way. We agree that the shedding of this parasite could be intermittent by carrier or infected snakes and regular testing is necessary to detect true positive animals. We have decided to include in our manuscript the fact that a negative PCR result after one fecal testing could provide false negative results, which further underlines the importance of caution for breeders and anyone in contact with such animals. Kindly see Lines 193-195 in the corrected version of the manuscript.

-. Cryptosporidium spp. is related to proliferative gastritis and rapid decreasing of body condition in snakes in captivity, particularly. Because no postmortem studies were performed here, the authors are recommended to include information about body condition of each of snakes tested here, and if another common disease (e.g. inclusion body disease) was tested in the way to check the health status of these wild-caught snakes.

Answer: Based on the information provided to us, none of the snakes exhibited any overt signs of disease or physical abnormalities, and no data regarding mortality were reported. To the best of our knowledge, no other analyses or professional examinations were performed on these individuals, apart from our testing of fecal samples for the presence of Cryptosporidium. We have decided to include these information in the corrected version of the manuscript, kindly see Lines 106-107.

Also, minor editing is suggested, as follows

-. The authors are recommended to include all common names of the tested snakes, together with scientific names of each one.

Answer: All common names of the tested snakes have been included in our manuscript; please see the “Snake species“ column in Table 1 in the corrected version.

-. The authors are recommended to unify the naming of the protozoal parasite studied here as "Cryptosporidium" or "Cryptosporidium spp." in all manuscript.

Answer: We unified the naming of the protozoal parasite studied here as Cryptosporidium spp. throughout the manuscript. We also used more accurate terms such as “cryptosporidial infection“ and “cryptosporidial oocysts“ where appropriate. Kindly see the corrected version of our manuscript."

-. The authors are recommended to include reference/s for the molecular study, if the the set of primers and PCR cycling conditions were obtained from previous published papers.

Answer: The reference has been added.

Round 2

Reviewer 1 Report

Comments and Suggestions for Authors

Dear authors,
I would like to thank you for diligently addressing the suggested changes and for implementing the necessary modifications to improve the manuscript. I have nothing further to add.

Best regards.

Author Response

Thank you.

Reviewer 3 Report

Comments and Suggestions for Authors

Dear authors

You addressed all comments and suggestions previously Made. I have no additional suggestions to provide. 

Author Response

Thank you.